# Studying Epigenetics of Cardiovascular Diseases on Chip Guide

**Bandar Ali Alghamdi [1], Intisar Mahmoud Aljohani [2], Bandar Ghazi Alotaibi [2], Muhammad Ahmed [3], Kholod Abduallah Almazmomi [2], Salman Aloufi [2],\* and Jowhra Alshamrani [2]**

1   Department of Cardiology and Cardiac Surgery, King Fahad Aramed Fornce Hospital, Jeddah 23311, Saudi Arabia; bandaralghamdi90@gmail.com

2   Department of Biotechnology, Taif University, Taif 21944, Saudi Arabia; entsarlafi@outlook.sa (I.M.A.); bandr123412@yahoo.com (B.G.A.); kholod.almazmomi@gmail.com (K.A.A.); alshmranijowhra@hotmail.com (J.A.)

3   Centre for Excellence in Molecular Biology, University of the Punjab, Lahore 54590, Pakistan; muhammad.ahmed@cemb.edu.pk

\*   Correspondence: s.aloufi@tu.edu.sa

**Abstract:** Epigenetics is defined as the study of inheritable changes in the gene expressions and phenotypes that occurs without altering the normal DNA sequence. These changes are mainly due to an alteration in chromatin or its packaging, which changes the DNA accessibility. DNA methylation, histone modification, and noncoding or microRNAs can best explain the mechanism of epigenetics. There are various DNA methylated enzymes, histone-modifying enzymes, and microRNAs involved in the cause of various CVDs (cardiovascular diseases) such as cardiac hypertrophy, heart failure, and hypertension. Moreover, various CVD risk factors such as diabetes mellitus, hypoxia, aging, dyslipidemia, and their epigenetics are also discussed together with CVDs such as CHD (coronary heart disease) and PAH (pulmonary arterial hypertension). Furthermore, different techniques involved in epigenetic chromatin mapping are explained. Among these techniques, the ChIP-on-chip guide is explained with regard to its role in cardiac hypertrophy, a final form of heart failure. This review focuses on different epigenetic factors that are involved in causing cardiovascular diseases.

**Keywords:** cardiovascular diseases; epigenetics; DNA methylation; histone modifications; microRNAs; ChIP-on-chip guide





## 1. Introduction

Epigenetics is defined as the study of inheritable changes in the gene expressions and phenotypes that occurs without altering the normal DNA sequence. These changes are mainly due to an alteration in chromatin or its packaging, which changes the DNA accessibility [1]. These epigenetic changes are often due to the interactions of the genes with their surrounding conditions or the environment, causing either an increase or a decrease in gene expression or potentially leading to gene silencing as in the case of obesity, diabetes, and hypertension [2].

Cardiovascular disease (CVD) epigenetics is considered to be a relatively new field. One of the leading causes of death worldwide, i.e., heart failure (HF), occurs when the myocardium undergoes functional and structural modifications. These processes result in the transcriptional and genomic reprogramming of cardiomyocytes and other neighboring cells [3]. Due to a lack of knowledge in comprehending complex CVD pathophysiology, scientists are searching for other pathways. Epigenetic modifications of the genome represent one such pathway. The mechanisms of epigenetics can be best explained via (Figure 1) DNA methylation, histone modification, and noncoding or microRNAs. They regulate gene expressions and affect the related risk factors, i.e., diabetes, hypertension, inflammation,

and atherosclerosis. Unlike other genetic aberrations and mutations, epigenetic modifications are dynamic and can be altered either by therapeutic approaches or by lifestyle [4].

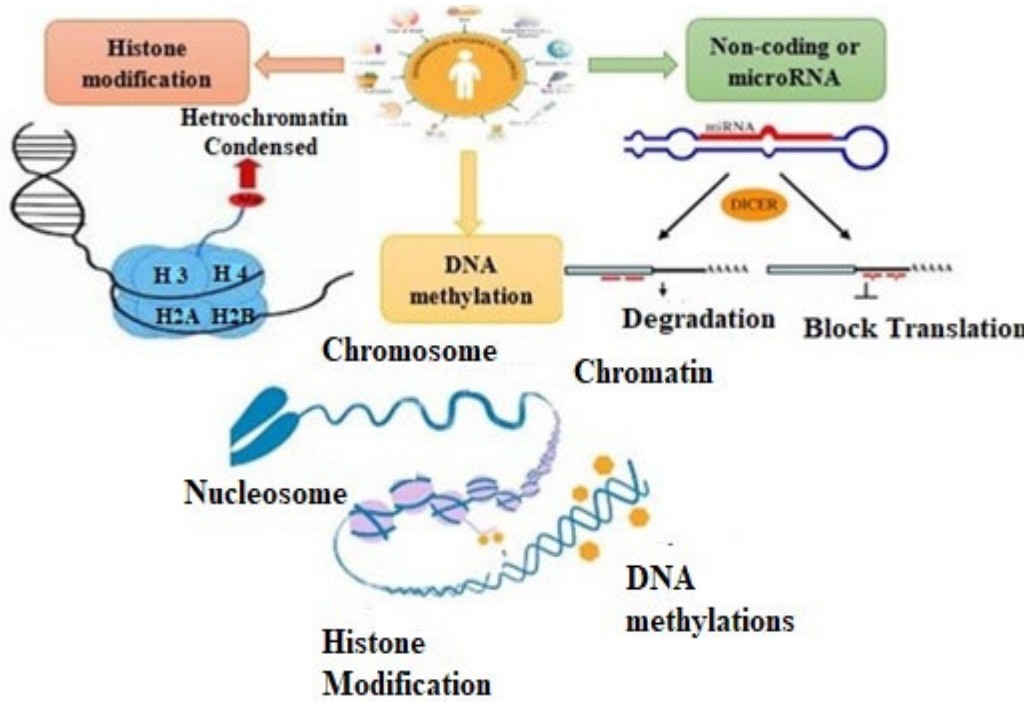

**Figure 1.** Epigenetics and environmental factors.

### 1.1. DNA Methylation

DNA methylation basically occurs at cystine residues in CpG sequences at the fifth position. These CpG sequences, instead of localizing in the coding region, localize in the promoter region [5]. This results in suppression of gene transcription either indirectly by recognizing the methylated sites using chromatin-modifying enzymes or directly by impeding the attachment of transcriptional factors to the promoter region in DNA [6]. DNA methylation is mediated by DNA methyltransferases such as DNMT3b, DNMT3a, and DNMT1. Methylation status during replication is maintained by DNMT1, which recognizes the hypermethylated DNA. On the other hand, DNMT3a and DNMT3b are involved in de novo methylation [7].

### 1.2. Histone Modification

Post-translational histone modification consists of phosphorylation, ubiquitination, acetylation, and methylation. These modifications occur in different patterns, regulating the shifting of the open chromatin structure (euchromatin) to a compact chromatin structure (heterochromatin) and vice versa [8]. Histone acetyltransferase (HAT), histone deacetylase (HDAC), histone methyltransferase (HMT), and histone demethylase (HDM) enzymes interact specifically at methylated DNA regions, thus causing gene transcription or repression [9,10].

### 1.3. Noncoding or MicroRNAs

Recently, it was discovered that noncoding RNAs are involved in gene regulation and genetic programming in both a healthy state and a CVD state [11]. It was found that 98% of the human genome, which undergoes transcription without encoding for proteins, produces noncoding RNAs that are involved in important structural and regulatory functions. On the basis of size, these noncoding RNAs are classified into two types, i.e., long noncoding RNAs with a size of 0.2 kb to 2 kb and small noncoding RNAs, which consist of endogenous short interfering RNAs, PIWI-interacting RNAs, and microRNAs. Studies have also shown that

these noncoding RNAs act as biomarkers of cardiovascular diseases (CVDs) and contribute to the pathogenesis of CVD [12].

## 2. Cardiovascular Epigenetics

Epigenetics or epigenomics shows a critical association between phenotypic expression and genomic coding that is affected by both environmental and genetic factors (Figure 2). Studies have shown that cardiovascular risk factors may affect and rearrange epigenetic patterns, and these cardiovascular biomarkers are said to be affiliated with epigenetic modifications. These epigenetic modifications are associated with clinical and subclinical cardiovascular diseases. Epigenetics is considered to be interconnected with genetics because these modifications (DNA methylation and histone modification) can change the expression of these genetic variations [13].

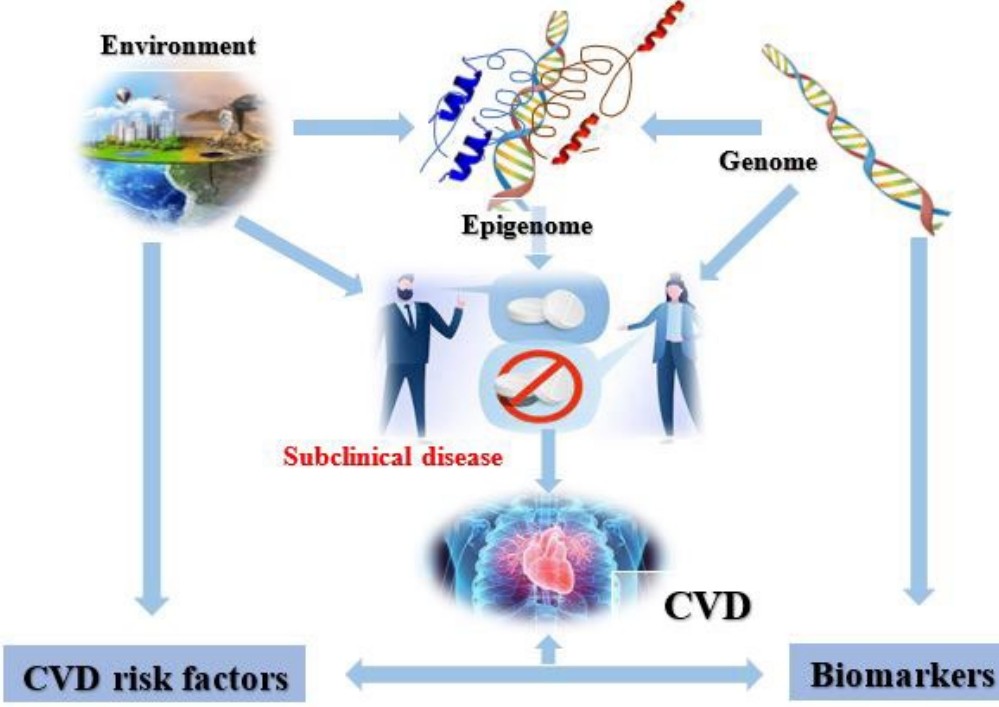

**Figure 2.** Epigenomics to cardiovascular diseases and risk factors.

## 3. Epigenetic Modifications in CVD

### 3.1. Role of DNA Methylation in CVD

Scientists have thoroughly studied the pathogenesis of cardiovascular diseases; however, there is still a need to explore the role of epigenetics. Table 1 summarizes the epigenetic modifications in cardiomyopathy. Using animal models for studies, scientists have demonstrated that DNA methylation plays an essential role in cardiovascular diseases and atherosclerosis [14]. Two important genes that are involved in DNA methylation are DNA methyltransferases (DNMTs) and methylene tetrahydrofolate reductase (MTHFR). A mouse model deficient of these two genes showed DNA hypomethylation [14,15]. The formation of aortic fatty-acid streaks was also shown by Chen et al. in an MTHFR mouse model. Furthermore, an enhanced expression of inflammatory mediators indicating hypomethylation was also detected in leucocytes of DNMT [16].

**Table 1.** Role of epigenetic modifications: DNA methylation and histone modification.

| Epigenetic Modification | Binding Domains | Targets | Phenotypes | Altered Gene Expressions | References |
|---|---|---|---|---|---|
| **DNA methylation** | None | CpG islands | Change in gene expression of angiogenic factors and heart failure | Up- or downregulation | [17,18] |
| **Histone modification** | | | | | |
| **Ribosylation** | PARP1 | HDACs, PARP1, histones, and brg1 | Enhanced fetal β-MHC expression, complex with Brg1 and HDACs, heart failure, and cardiac hypertrophy | Upregulation | [19] |
| **Phosphorylation** | PKD, AMPK, JAK2, Rsk2, Aurora | HDACs, H2B, H4Y41, S28/S10, H3 | Transcriptional activation, cardiac hypertrophy regulation, cellular proliferation, and mitotic activity | Up- or downregulation | [20–22] |
| **Deacetylation** | HDAC Class II (9,5,4) | Tails of histones | Cardiac hypertrophy negative regulation and inhibition of MEF2 activity (myocyte enhancer factor 2) | | [23,24] |
| **Acetylation** | P300, CBP (CREBP-binding protein) | K19, K16, K12, K8, H4K5, H3K4 | Regulation of cardiac hypertrophy | Upregulation | [25,26] |
| **Demethylation** | UTX, JMJD2A | H3K27me3, H3K36me3, H3K4me3 | Embryo lethality, heart malformation, and stimulation of cardiac hypertrophy | Up- or downregulation | [27,28] |
| **Methylation** | DOTIL, PTIP | H3K79me, H3K27me3, H3K9me3 H3K4me3, H3K4me2 | Dilated cardiomyopathy, heart failure and angiogenesis, and activation of fetal cardiac gene | Up- or downregulation | [29–31] |

Dietary folate and vitamin levels also affect the DNA methylation status, and their supplementation to female mice before conception resulted in increased CpG methylation in offspring. This led to characteristic phenotypes in offspring such as lengthened lifespan and reduced susceptibility to obesity and insulin resistance [14].

A decrease in DNA methylation was represented in the aortas of ApoE knockout mice, which was detected after 4 weeks; any histological alterations detected were associated with atherosclerosis [32]. In atherosclerotic tissue, estrogen receptors α and β showed enhanced methylation in the promoter region. This was due to hypermethylation of the HSD11B2 gene and reduction in the global genomic methylation of blood leukocytes of hypertension patients [33]. A prevalence of elevated Alu methylation status in peripheral

blood leucocytes was found in Chinese people, which was associated with the obesity and cardiovascular diseases [34].

### 3.2. Role of Histone Modification in CVD

Histone modifications in association with microRNAs and DNA methylation are considered a dynamic process involved in the modulation of gene expression and chromatin remodeling. The main component histones of the nucleosome, i.e., H2A, H2B, H3, and H4, are modified via post-translational modifications such as biotinylation, sumoylation, acetylation, phosphorylation, ADP ribosylation, and methylation [8,35] (Table 2).

Histone acetylation and deacetylation is carried out by HATs and HDACs as mentioned above. Studies have shown that there are certain types of acetyltransferases for histone acetylation that are involved in deletion of lethal genes from embryo and heart development, i.e., CREBP (cAMP responsive element-binding protein) or p300 [36]. If the HAT domain in CREB or p300 is ablated, it may lead to abnormality in the cardiovascular system [37]. Transcriptional factors such as GATA4 when acetylated by p300 or overexpressed may lead to depressed cardiac dilation in the murine heart [38,39].

There are certain cell-specific histone modifications that control eNOS expression (nitric oxide synthase) in endothelial cells, which are considered to be essential in vascular functioning. It was found that there is a high amount of acetylated H4K12 and H3K9 in the eNOS gene core promoter in endothelial cells. Furthermore, in the case of cardiac degeneration, there is remarkable decrease in the expression of eNOS [40]. Further studies revealed that the expression of eNOS gene can also be controlled by methylation in the promoter region of this gene, i.e., H3K4me3 and H3K27me3. A reduction in angiogenesis, which is triggered by hypoxia, is promoted by the enhanced expression of JMJD3 (histone demethylase), due to an increase in the ratio of active H3K27me3 to H3K4me3 [41].

Histone methylation basically occurs at lysine or arginine residues with mono/di/trim ethylated histones (H3Kme3) at H4K20, H3K36, H3K27, H3K9, and H3K4 [19]. Unlike acetylation, which activates chromatin, methylation can result in a poised or repressed state of chromatin. Moreover, histone methylation is involved in not only heart development but also cardiac hypertrophy and heart failure [42,43]. It was revealed in a study that myocardial stress was observed in cardiac failure and hypertrophy due to transcriptional reprogramming and chromatin remodeling. Furthermore, in hypertrophic and cardiomyopathic mouse models, there was an increase in the expression of the Brg1 gene (Brahma-related), whereas a decrease in hypertrophy was observed in models with reduced expression of the Brg1 gene [44].

### 3.3. Role of microRNA in CVD

Recently it was discovered that microRNAs are involved in epigenetic mechanisms that can cause cardiovascular degeneration and diseases (Table 3). Elevated levels of miR-127 found in patients of atherosclerotic plaque can disrupt the endothelium by inhibiting SIRT1. This results in destruction of the vascular senescence [45]. In myocardial infarction patients and developed mouse models, miR-499 and miR-133b were upregulated and found to be potential candidates for cardiovascular disease biomarkers [46]. Moreover, patients with coronary artery disease exhibit significantly decreased levels of miR-145 and miR-126 [47,48]. Unstable angina patients display elevated levels of miR-370, miR-198, and miR-134, which can be life-threatening and lead to cardiovascular diseases [49]. Higher levels of miR-624 and miR-340 were also reported by Sondermeijer and his colleagues in cardiovascular disease patients [50].

**Table 2.** Epigenetic modifications: role of microRNAs.

| Micrornas | Methodology | Site of Expression | Fold Expression | References |
|---|---|---|---|---|
| **Heart failure** | | | | |
| **miR-212, -129, -21** | Array | Human heart tissue | >1.5× upregulated | [51] |
| **miR-133b, -92** | | | downregulated | |
| **miR-342, -214, -181b, -125** | Array | Left heart ventricle | Upregulation | [52] |
| **miR-497, -139, -125b** | qRT-PCR | PBMCs | >2× downregulation | [53] |
| **miR-29b** | | | >2× upregulation | |
| **miR-214** | Bead-based hybridization | Left heart ventricle | 2–2.7× downregulation | [54] |
| **miR-24, -214, -125b, -195** | Northern blots | Left heart ventricle | 1.3–3× upregulation | [55] |
| **Coronary artery disease** | | | | |
| **miR-624, miR-340** | Array | Platelets | 1.5× upregulation | [50] |
| **miR-147** | qRT-PCR | PBMCs | 4× downregulation | [56] |
| **miR-370, -134** | | | 3.1–12× upregulation | |
| **miR-499, -126-133a** | qRT-PCR | Plasma | 2–20× upregulation | [57] |
| **miR-208a, -133** | Array | Plasma | Upregulation | [48] |
| **miR-17-92** | | | Downregulation | |
| **Acute myocardial infarction** | | | | |
| **miR-328** | qRT-PCR | Plasma | Upregulation | [58] |
| **miR-306** | Array | PBMCs | Upregulation | [59] |
| **miR-1291** | | | Downregulation | |
| **miR-423-5p** | qRT-PCR | Plasma | 3–10× upregulation | [60] |
| **miR-375, miR-122** | qRT-PCR | Plasma | Downregulation | [61] |
| **miR-21** | qRT-PCR | Rat myocytes | Border cells: upregulation, infarcted cells: downregulation | [62] |
| **miR-223** | qRT-PCR | Plasma | Downregulation | [63] |

**Table 3.** Techniques for epigenetic studies [64].

| Technique | Abbreviation | Description |
|---|---|---|
| **Whole-genome bisulfite sequencing** | WGBS | This is an NGS technique used to evaluate the status of DNA methylation of cytosine residues across the genome and to directly determine the ratio of methylated molecules. DNA samples are treated with sodium bisulfite that only converts unmethylated cytosine into uracil and leaves the methylated cytosine unchanged. |
| **RNA sequencing** | RNA seq | This is used for determination of the cellular transcriptome. It allows the evaluation of differences in gene expression, mutations or SNPs, gene fusion, post-transcriptional modification, and alternative gene sliced transcripts using different treatments or groups. Along with mRNA transcripts, this technique also determines ribosomal profiling and different RNA populations, including small RNAs such as tRNA and miRNA, as well as total RNA. |

**Table 3.** *Cont.*

| Technique | Abbreviation | Description |
|---|---|---|
| **ChIP-on-chip** | ChIP-on-chip | This is a technology that combines two techniques, i.e., DNA microarray (chip) and chromatin immune precipitation (ChIP). It determines the in vivo interaction between DNA and proteins. |
| **Epigenome-wide association studies** | EWAS | This is used to determine the connection between specific identifiable traits or phenotypes in large human cohorts and a genome-wide set of epigenetic biomarkers. It determines whether there is an actual correlation between epigenetic perturbation and given phenotype. |
| **Genotype–Tissue Expression project** | GTEx | This project is used to provide valuable understanding regarding the mechanisms of gene regulation using existing knowledge of human gene regulation and expression in multiple tissues, not only from healthy subjects but also from various human diseases. |
| **Assay for transposase-accessible chromatin sequencing** | ATAC-seq | This is an HTS technology that provides access to a genome-wide map of chromatin. It provides specific information regarding genome-wide positions of the following: 1. Information on chromatin state annotation 2. Nucleosomes in regulatory regions 3. Transcriptional binding factors 4. Open chromatin |

From animal studies, scientists determined the role of miR-21 in the early phase of MI. It was found that the expression of miR-21 decreased in the infarcted area in comparison to its surroundings due to a left-ventricular coronary artery ligation created by acute MI [56]. Olivieri et al., on the other hand, tried to discover the diagnostic potential of microRNAs and discovered elevated levels of miR-423-5p in patients of congestive heart failure compared to those with MI. However, there was upregulation of miR-499-5p in both HF and MI [65]. Additionally, miR-499-5p was found to be a more sensitive marker in elderly patients compared to troponin T, thus helping to differentiate acute congestive heart failure from MI [65]. MicroRNAs that were isolated from whole blood were evaluated for prognostic and diagnostic properties, such that they could be used to predict MI in cases where ischemic heart disease biomarkers and troponin T were negative [60].

## 4. CVD Risk Factors

Over the last few decades, studies have started to link epigenetic factors such as atmospheric pollutants, diet, urban noise, smoking, and economic, social, and cultural circumstances to cardiovascular disease risk factors such as diabetes, aging, and hypertension in humans. Examples of such associations are given below.

### 4.1. Diabetes Mellitus

One of the major risk factors for cardiovascular disorders is diabetes mellitus, which is caused by both genetic and environmental factors. When considering external influences and integrating the DNA code, epigenetic modifications were identified as the cause [66]. It was found that insulin production is inversely related to DNA methylation of the promoter region of the insulin gene. Insulin secretion is affected by demethylation of the mature insulin-producing cells. Moreover, with respect to environmental influences, pathological diabetic factors may result in the development of diabetes [67]. It was also found that, in the 11p15 genomic region, a differentially methylated CCCTC-binding factor (CTCF) binding site is associated with type 2 diabetes at the population level [68].

T2D epigenetics helps in determining the destructive effects of hyperglycemia, i.e., **hyperglycemic or metabolic memory**, despite optimum control. Epigenetic signatures in inflammatory and pro-oxidant gene promoters are involved in atherosclerotic features, endothelial dysfunction, retinopathy, and diabetic nephropathy, despite normoglycemia being restored [69]. In peripheral blood mononuclear cells (PBMCs) from T2D patients after glycemic control, there was reduced methylation of adaptor p66$^{Shc}$ promoter (mitochondrial oxidative stress), which was found to be related to oxidative stress levels and endothelial dysfunction [70]. Recently, several specific epigenetic factors present on the histone 3 of type 2 diabetic patients, i.e., H3K4me, were discovered [71]. This epigenetic pattern is involved in the activation of NF-κB after its activation by methyltransferase Set7, which results in overexpression of pro-atherosclerotic genes such as VCAM-1, MCP-1, ICAM-1, COX-1, and iNOS [72]. Peripheral blood monocytes of patients with coronary artery disease and insulin resistance showed a decreased expression of histone 3 deacetylase SIRT1, thus showing the link among atherosclerosis, metabolism, and longevity [73].

### 4.2. Hypoxia: A Pathophysiological Factor

Another CVD risk factor is hypoxia (unavailability of required oxygen amount to maintain normal homeostasis), which is marked by a decline in transcriptional activity, resulting in upregulation of ubiquitous repressive histone methylation [74]. In order to maintain constitutive transcriptional activity of the gene encoding eNOS, histone modification is considered essential. During hypoxia, proximal promoter histones are acetylated, which results in a decrease in transcription with deadly effects on vascular homeostasis [75]. Additionally, histone deacetylase 3 was found to be essential for the survival of endothelial cells and development of atherosclerosis as a result of distributed blood flow at vessel bifurcation [76].

### 4.3. Aging

In aging, changes in epigenetic patterns called epigenetic drift result in a reduction in global DNA methylation. In the average blood, global DNA methylation showed a longitudinal decline in repetitive sequences, i.e., LINE-1 and Alu, in a normative aging study [77]. Aged stem-cell GWASs (genome-wide association studies) showed hypermethylation of genes involved in differentiation and hypomethylation of self-renewal gene promoters [78]. An increase in age marks a decrease in trimethylation of histone3 at lysine 27 and 9 (H3K9me27 and H3K9me3) and acetylation at lysine 9 (H3K9Ac). This leads to defective vascular repair and hematopoietic stem-cell dysfunction [79]. In aging, enhanced miRNA expression leads to suppression of PTMs (post-transcriptional modifications) of target genes and changes in endothelial functions. Cardiac and vascular function decline in elders is associated with derailed expression of miR-146, miR-217, miR-34a, and miR-29 [80]. Angiogenic potential may be decreased by an age-dependent decline in many long noncoding RNAs, i.e., Meg3, MIAT, MALAT-1, and ANRIL [12].

### 4.4. Dyslipidemia

The blood lipid profile (BPL) is affected by both environmental and genetic factors [81]. Epigenetic modifications of gene promoters involved in lipid and glucose metabolism contribute to atherogenesis. A study showed that participants that were exposed to the 1944–1945 famine with insulin resistance and obesity in adulthood showed alterations in blood methylation of leptin (lep) and insulin-like growth factor 2 (IGF-2) [82]. A study revealed that CPT1A methylation was associated with the triglycerides and very low LDL (low-density lipoprotein) cholesterol [83]. Furthermore, it was revealed that miR-33a/b is involved in the post-transcriptional regulation of insulin signaling and lipid metabolism, and inhibition of miR-33a/b may result in a decline in atherosclerosis via an increase in blood HDL (high-density lipoprotein) levels [84].

## 5. Cardiovascular Diseases and the Epigenome

### 5.1. Coronary Heart Disease (CHD)

Alterations in the DNA methylation levels of target genes and systemic and endothelial inflammation were found to be involved in CHD pathophysiology. This leads to rupture and destabilization of the atherosclerotic plaques, causing acute cardiovascular events [85,86]. Recent studies have shown the role of DNA methylation in equilibrizing the conventional predictors of cardiovascular risks [87]. A correlation was determined between cardiac computed tomography angiography (CCTA) features and blood-based methylation levels involved in the specific CGI-regulating HLA-G gene that encodes anti-inflammatory substances with immunomodulatory properties in obstructive CHD patients as compared to nonobstructive CHD patients [88]. The positive correlation between coronary calcium score and hypomethylation of the specific CGI-related HLA-G gene fragment disclosed that methylation was involved not only in predicting the severity of the disease but also as a noninvasive biomarker, thus leading to an improved CCTA prognostic value [88]. To identify the disease modules and blood-based differentially methylated regions (DMRs) associated with CHD incident events, the Comb-p and WGCNA algorithms were applied in two independent cohorts (replication sets: 2726 subjects; discovery sets: 2129 women) [89]. This study led to the identification of two modules highly enriched for immune-related processes and development processes. A positive correlation was determined with TG, highly sensitive C-reactive protein (hsCRP), and body mass index (BMI) after multivariate analysis [89].

### 5.2. Pulmonary Arterial Hypertension or PAH

Pulmonary arterial hypertension is defined as an incurable and rare disease which is characterized by consequent elevated pulmonary artery pressure and vasoconstriction due to three major endophenotypes, i.e., inflammation, cell migration/proliferation, and endothelial dysfunction. PAH is triggered by the association between epigenetic and genetic risk factors when exposed to detrimental environmental factors [90,91]. Interaction and transcriptomic profiling of pulmonary arterial endothelial cells (PAECs) obtained from late-stage PAH patients and normal controls during the time of lung transplant helped in the construction of an integrated regulatory network by integrating chromatin [92]. This resulted in thorough remodeling of agile enhancers regulated by specified transcriptional factors and marked by H3K27ac, which provoked perturbation of endothelial-to-mesenchymal transition and angiogenesis processes in response to specific growth factors signals in PAECs for targeted genes such as eNOS3 [92].

## 6. Epigenetic Dysfunctional Responses in CVD

During early adaptive responses to cell injury, sensitive epigenetic molecular networks become abnormal after establishing chronic stress in the heart. Our focus is on the main phenotypes, i.e., mitochondrial dysfunction and fibrosis.

### 6.1. Mitochondrial Dysfunction

Higher circulating mtDNA can lead to a poor prognosis of HF patients and left-ventricle (LV) remodeling [93]. mtDNA gene methylation upregulates protease expression and silencing of survival pathways, which triggers cardiomyocyte death [94]. A study revealed hypermethylation in four mitochondrial genes of CVD patients, i.e., mitochondrially encoded tRNA leucine 1 (UUA/G) (MT-TL1) and cytochrome-c oxidases I, II, and III (MT-CO1, MT-CO2, MT-CO3) [94].

### 6.2. Fibrosis

During remodeling of the left ventricle (LV), there is an increase in collagen, which occupies the area between vessels and myocytes. This results in progression of reparative fibrosis by effecting diastolic ventricular function and tissue stiffness [95]. Some studies reported that global DNA hypermethylation was induced by increasing the level of hypoxia

via upregulation of DNMT3B and DNMT1. This may result in overexpression of alpha-smooth muscle actin ($\alpha$-SMA) and collagen 1 genes. Reduced $\alpha$-SMA and collagen 1 expression was observed in human primary cardiac fibroblasts by siRNA administration to inhibit DNMT3B expression, which suggests a crucial putative drug target [96].

## 7. Techniques for Individual Epigenetic Mapping of Cardiovascular Disorders

In contrast with the belief that every individual has a somewhat similar genome, scientists proposed that all the epigenomes are not born same. Instead of an exception, diversity is considered to be a norm when it comes to the epigenetic modifications. When scientists completed the Human Genome Project, it became the cornerstone of genomic research [97]. NGS, advances in bioinformatics, and GWAS helped in analyzing the increasing number of genomic datasets [98]. However, despite these advances, the field of precision medicine in epigenomic cardiology remains uncharted. Although there are some barriers left to overcome, efforts have begun to complete the cardiovascular epigenome [99].

In order to enable clinical applications, various technological advancements have been introduced in the field of epigenomics. However, there still remains a key challenge for cheap performance of whole-genome bisulfite sequencing (WGBS) studies for EWAS (epigenome-wide association studies) (Table 4, Figure 3).

**Table 4.** Comparison of different scChIP seq methods [100].

| Methods | Strategy | Mapping Rate | Cell State | Device for Cell Sorting |
|---|---|---|---|---|
| CUT and Tag | ChIP-free, Tn5-barcoding (1 round) | 97% | Native | Costly Takara ICELL8 |
| Co-BATCH | ChIP-free, Tn5-barcoding (for 2 rounds) | 94% | Fixed and native | FACS |
| sc-itChIP-seq | ChIP and Tn5-barcoding (1 round) | 94% | Fixed and native | FACS |
| scDrop-ChIP | Microfluidic system and ChIP for droplet formation | 70% | Native | Costly microfluidic device |

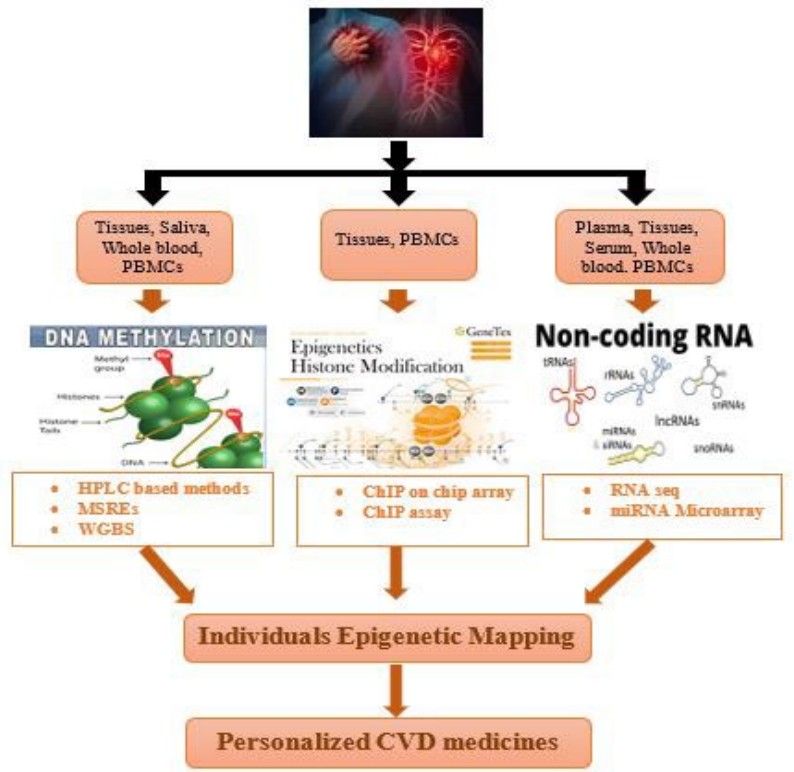

**Figure 3.** Application of epigenetic mapping techniques for CVD patients.

## 8. ChIP-on-Chip Guide

Chromatin immunoprecipitation (ChIP) followed by DNA microarray (chip), collectively called ChIP-on-chip, was the earliest technology used for large-scale epigenetic mapping, allowing the scientists to identify the protein–DNA interactions on a genome-wide level [101]. It is based on the principle of DNA microchip hybridization, where a large number of probes covering a specific region or whole genome are seeded on a high-density chip. However, this technique suffered disadvantages such as signal bias, low resolution, inapplicability to a broad range of species, and ambiguous factors introduced by probe design [102].

Chromatin immunoprecipitation sequencing (ChIP-seq) techniques, in comparison to ChIP-on-chip techniques, provides greater coverage, less noise, and higher resolution [103,104]. Due to the fast decrease in the price of second-generation sequencing (SGS), ChIP seq has now become an essential technology in determining the epigenetics and gene regulation. This technology can also be used to determine enhancers, transcriptional factors, and various other regulatory elements [105].

### 8.1. Traditional ChIP Seq

For protein–DNA complexed, ChIP seq procedures are performed to enrich the DNAs that are attached to specific proteins. It is a multistep experiment. At first, DNA is crosslinked with proteins via formaldehyde. Then, this crosslinked complex is subjected to sonication, which breaks it down into 200–600 bp small fragments. Then, the protein–DNA complex of interest is immunoprecipitated using an antibody against the respective protein. This releases the DNA, which is then subjected to DNA end repair, ligation of the adapter molecule, and construction of the library. Next, the required DNA is sequenced (Figure 4) [106].

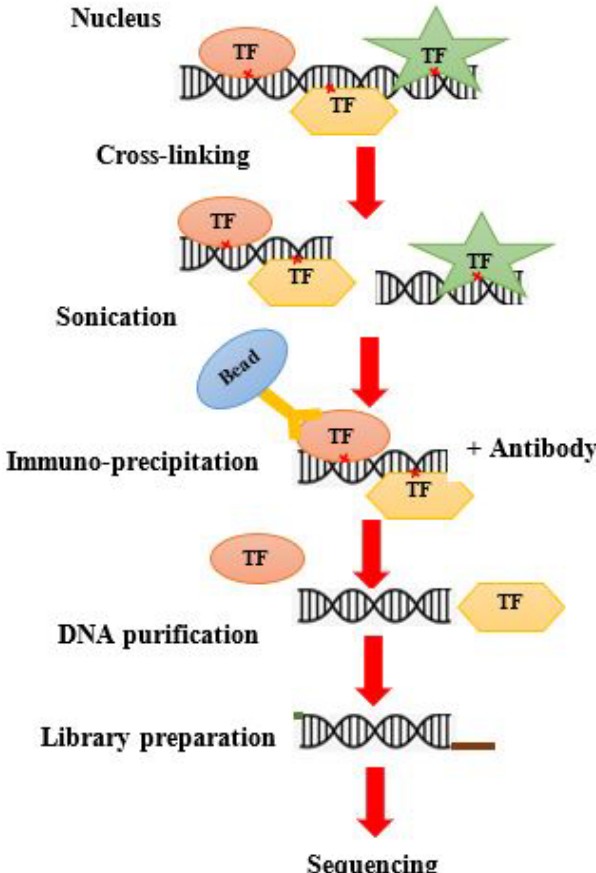

**Figure 4.** Workflow of ChIP seq.

*8.2. Single-Cell ChIP Seq*

Traditional ChIP seq technology is unable to identify the chromatin signature of individual cells. For this purpose, a new technique was introduced that was able to determine genetic diversity in heterogenous cell populations and obtain knowledge regarding the evolution of tumor population; this technique is called single-cell ChIP (scChIP). Droplet-based scChIP (Drop ChIP) combines scDNA barcoding with a microfluidic device, allowing scientists to gain a comparatively low map coverage per cell [107,108]. scChIP seq technology allows the clustering of cell populations on the basis of chromatin landscape diversity and determination of unique chromatin features of the population; for example, the decline in H3K27me3 biomarkers in some cells may lead to chemoresistance or can cause CVD [100]. Table 4 shows the comparison of different ChIP seq methods and the mapping percentage.

## 9. Role of ChIP Seq in Determining Epigenetic Signature Underlying Cardiac Hypertrophy

One of the leading causes of mortality worldwide is heart failure (HF), which is frequently followed by a condition called cardiac hypertrophy, a condition in which there is an expression of genes that are only activated during the fetal stage and a repression of genes that need to be activated in adults [109]. Although epigenetics is considered to be essential in regulating transcription, its role remains unknown in cardiac hypertrophy. Genome-wide association studies (GWASs) of histone 3 lysine-36 trimethylation (H3K36me3) and DNA methylation in normal hearts and cardiomyopathic hearts of humans divulged a wide range of epigenetic patterns [110]. When idiopathic dilated cardiomyopathic patients were interrogated with cardiac methylome, differences in methylation were detected not only in heart disease-related pathways but also in genes of heart failure with yet unknown functions such as adenosine receptor A2A (ADORA2A) and lymphocyte antigen 75 (LY75) [111]. High levels of miRNA-508-5p and miR-499 and low levels of circulating miRNA-342-3p, miRNA-30b, miRNA-142-3p, and miRNA-103 were detected in patients with advanced heart failure [112,113]. Moreover, if low levels of miRNA-423-5p remain in these patients for a long time, it may result in bad outcomes [114].

A study was conducted in adult mouse cardiomyocytes to describe epigenetic changes that occurred when they were subjected to a pro-hypertrophy stimulus in vivo. Genome-wide chromatin maps were generated and compared for the gene expression of normal and hypertrophic cardiomyocytes. These cardiomyocytes were isolated from the left ventricle of mouse hearts, which were subjected to transverse aortic constriction, and ChIP seq was performed using these cells with antibodies against active regulatory regions associated with three markers, i.e., H3K4me3, H3K27ac, and H3K9ac, as well as repressed regions represented by H3K27me3, H3K9me3, and H3K9me2 [115,116], and transcribed genes represented by H3K4me3 [117].

Overall, 9.1% of the genome of cardiac hypertrophic cells showed an alteration in the distribution of at least one histone mark was rearranged to transcriptional start sites (TSSs). Promoters of hypertrophic cardiomyocytes showed distinguished epigenetic patterns, and 9207 active enhancers were discovered with modulated activity. A role for myocyte enhancer factor (MEF)2C and MEF2A in regulating enhancers was identified by analyzing the transcriptional network within which the genetic elements tried to orchestrate hypertrophy gene expression [118].

## 10. Conclusions

During the past few decades, thorough studies were conducted to uncover the molecular mechanisms involved in chromatin regulation and conditions of diseases that arise from the epigenetic misregulation. The regulatory mechanisms that control the establishment of chromatin domains and their conserved nature suggest that dramatic changes that occur in gene expression arise from an alteration in the chromatin landscape. This is the case for most overlapping modification layers that regulate transcription processes in the epigenetic context. In the case of disease conditions in adults, due to the large variation in the

phenotypes between patients, a more complex role is displayed by the epigenetic changes. Changes in the epigenetic factors related to the respective disease suggest that these abnormal proteins undergo alterations in subunit performance and lack biological repetition. Surprisingly, it was discovered cardiomyopathy genomics in human patients were very sensitive to these epigenetic aberrations. This suggests that the type of epigenomic factor expressed during heart formation is tissue-specific. Thus, this represents a new avenue to study these changes in molecular factors. Even today, scientists have failed to fully explore tissue heterogeneity in the setting of CVD. It needs to be determined whether methylation alterations n circulating leukocytes may disclose myocardial processes, thus providing accurate and reliable biomarkers.

**Author Contributions:** All authors have equal role. All authors have read and agreed to the published version of the manuscript.

**Funding:** This research received no external funding.

**Institutional Review Board Statement:** Not applicable.

**Conflicts of Interest:** The authors declare no conflict of interest.

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
