# Peer review of "Studying Epigenetics of Cardiovascular Diseases on Chip Guide"

_cardiogenetics, doi:10.3390/cardiogenetics12030021_

Round 1

Reviewer 1 Report

The authors submitted a narrative review in which they tried to elucidate different epigenetic factors that are involved in causing cardiovascular disease. The aim of the article is clear and concise. The manuscript has a logical structure and several colore figures. The tables are clear and legible. The conclusive part covers ever aspect of the study . The article is well referenced and written. However, I would like to put forward several isuues to discuss.

1. Section Introduction. The authors should avoid reporting figure with estblished opinion on the topic. Yet, the section should point at novelity, but not well-known evidence a description of which seems to be superfluous.

2. Methodology of the study. Albeit this is no systematic review, thet authors should describe the methodology of searching the articles, evaluating their quality and the role of each author in it.

3. The authors mixed CV risk factors, co-morbidities and CVD in the subsections of the paper, Please, give a clear description of epigenetic changes having clinical significance separately for each point.

4. I see no point of predictive models and targets for management of CVD based on epigenetic modifications of various molecules in the article. For instanse, metabolic memory phenomenon is well known target for T2D therapy, but it is not reported. Yet, lipid toxicity along with mitachondrial dysfunction seem to be under question of the article.

5. Please, extend the section Conclusion, so that it covers practcically useful aspects and underlines the direction for the future.

Author Response

 I am much obliged to you for your kind time to review my data. kindly check the attachments.Corrections according to comments also have been added to new menuscript. 

Comment 1:

Introduction is improved by adding new materials.

Cardiovascular diseases (CVD) epigenetics are considered to be a relatively new field. One of the leading causes of death worldwide i.e. heart failure (HF) occurs when myocardium undergoes functional and structural modifications. These processes results in the transcriptional and genomic reprogramming of cardiomyocytes and other neighboring cells. Due to lack of knowledge to comprehend complex CVD pathophysiology, scientists are searching for other pathways. Epigenetic modifications of the genome are one of these pathways. The mechanisms of epigenetics can be best explained via DNA methylation, Histone modification and non-coding or micro-RNAs. They regulate gene expressions and affects the related risk factors i.e. diabetes, hypertension, inflammation and atherosclerosis. Unlike other genetic aberrations and mutations, epigenetic modifications are dynamic and can be altered either by therapeutic approaches or lifestyle.

Comment 2:

The data is taken from different research articles and is used as literature survey. Then data is combined and each section covers its meaning. The best possible results are added in the article.

Comment 3:

The sections have been separated and each section covers its clinical significance now.

Section 3: It explains role of DNA methylation, histone modification and different miRNA molecules in the CVD epigenetics.

Section 4: It explains CVD risk factors and their epigenetics. It covers:

  • Diabetes
  • Hypoxia (a pathophysiological factor)
  • Ageing
  • Dyslipidemia

Section 5: It explain different CVDs and their epigenomes. This section covers:

  • PAH
  • CHD
  • CdLS
  • DiGeorge syndrome

Comment 4:

Metabolic memory is now added as a T2D target.

T2D epigenetics helps in determining the destructing effects of hyperglycemia i.e. hyperglycemic or metabolic memory despite optimum control. Epigenetic signatures at inflammatory and pro-oxidant gene promoter are involved in atherosclerotic feature, endothelial dysfunction, retinopathy and diabetic nephropathy though normoglycemia is restored. In peripheral blood mononuclear cells (PBMCs) from T2D patients after glycemic control, there is reduced methylation of adaptor p66Shc promoter (mitochondrial oxidative stress) and is found to be related with oxidative stress levels and endothelial dysfunction. Recently several specific epigenetic factors present on the histone 3 of the type II diabetic patients i.e. H3K4me were discovered. This epigenetic pattern is involved in activation of NF-κB after it is activated by methyltransferase Set7 which results in overexpression of pro-atherosclerotic genes like VCAM-1, MCP-1, ICAM-1, COX-1 and iNOS. Peripheral blood monocytes of the patients with coronary artery disease and insulin resistance showed decrease in expression of histone 3 deacetylase SIRT1 thus showing link between atherosclerosis, metabolism and longevity

Moreover, epigenetic dysfunctional responses in CVD have been added covering mitochondrial dysfunction and fibrosis.

Epigenetic dysfunctional responses in CVD:

During early adaptive responses to cell injury, epigenetic sensitive molecular networks become abnormal after establishing chronic stress in heart. Our focus is on main phenotypes i.e. mitochondrial dysfunction and fibrosis.

Mitochondrial dysfunction:

 Higher circulating mtDNA can cause poor prognosis of HF patients and left ventricle (LV) remodeling. mtDNA genes methylation upregulates proteases expression and silencing of survival pathways which triggers cardiomyocytes death. A study revealed hypermethylation in 4 mitochondrial genes of CVD patients i.e. mitochondrially-encoded tRNA leucine 1 (UUA/G) (MT-TL1), cytochrome-c-oxidase I, II and III (MT-CO1, MT-CO2, MT-CO3).

Fibrosis:

During remodeling of left ventricle (LV), there is an increase in collagen which occupies the area between vessels and myocytes. It results in progression of reparative fibrosis by effecting diastolic ventricular function and tissue stiffness. Some studies reported global DNA hypermethylation was induced via increasing level of hypoxia by upregulation of DNMT3B and DNMT1. It may result in overexpression of alpha-smooth muscle actin (α-SMA) and collagen 1 genes. Reduced α-SMA and collagen 1 expression was observed in human primary cardiac fibroblasts by siRNA administration to inhibit DNMT3B expression which suggests a crucial putative drug target.

Comment 5:

Conclusion is now edited.

Conclusion:

During the past few decades, thorough study was done to uncover the molecular mechanisms that are involved in chromatin regulation and conditions of diseases that arise from the epigenetic mis-regulation. The regulatory mechanisms that controls the establishment of chromatin domains and its conserved nature suggested that dramatic changes that occurred in gene expression occurred due to alteration in the chromatin landscape. This has been the case for most of the part in context of overlapping modification layers that regulate transcription process and epigenetic context. In case of disease conditions in adults, due to large variation in the phenotypes between the patients a more complex role is displayed by the epigenetic changes. Changes in the epigenetic factors that are related to the respective disease suggest that these abnormal proteins have undergone alterations in its subunit performance and that they lack biological repetition. Surprisingly it was discovered that to these epigenetic aberrations, cardiomyopathy genomics in human patients were found to be very sensitive. This suggests that the type of the epigenomic factor that are expressed during the heart formation are found to be specific in that tissue. Thus, it becomes new avenue to study these changes at molecular factors. Even today, scientists failed to fully explore tissue heterogeneity in setting of CVD. It is needed to be determined whether methylation alterations n circulating leukocytes may disclose myocardial processes to provide accurate and reliable biomarkers.

Reviewer 2 Report

The topic of the article seems interesting, but the article itself needs to be seriously corrected.

Comments:

1.      The quality of the figures should be improved. For example, in figure 1 the text captions are poorly readable, and the images themselves are poorly visible. In figures 1-2 there is a transfer of text between the lines, which should not be the case. In the captions to the figures it is necessary to give the abbreviations (for example figure 3 has a lot of abbreviations).

2.The correctness of some sentences needs to be checked, e.g. «It was found that elevated levels of mir127 in patients suffering from atherosclerotic plaque can disrupt the endothelium which ultimately destructs the vascular senescence by inhibiting SIRT1» .

3. It must be clarified what is meant by "cardiovascular degeneration", as degenerative diseases are not considered separately.

4. It is necessary to check that abbreviations such as "histone deacetylases (HDM)" are correct. All abbreviations must be written in full when first mentioned.    

5. The structure of the article should be improved. It is recommended that sections be numbered.

6.      The section "Role of cardiovascular risk factors and epigenetics in cardiovascular disease" needs to be structured, as it contains diseases (Diabetes mellitus, Coronary heart disease), pathophysiological conditions (Hypoxia), and the section Heart failure includes different nosological units, including cardiomyopathies. It is also recommended to change the name of the section, as Heart failure is not a risk factor, but is the final outcome of many cardiovascular diseases.

7. Phenotypes of patients, including those with cardiomyopathies, are discussed in the conclusion, but the main body of the text does not address the phenotyping of patients with cardiovascular disease and the role of epigenetics in it.

8.      Much of the literature is older than 5 years, especially in the first part of the list; adding newer sources is recommended.

9.      It is recommended to consider combining Tables 1 and 2.

Author Response

I am much obliged to you for your kind time to review my data. kindly check the attachments.Corrections according to comments also have been added to new menuscript and comments have answers separately.

Round 2

Reviewer 1 Report

The authors submitted a revised verson of the paper along with a comprehensive explanation of the ways by which these corrections were made. I have no serious flaws to the artcle in its revised version.

Author Response

I am much thankful to you for your kind reviews. 

Reviewer 2 Report

The authors have done a lot of work and made many important corrections. However, some comments still remain:

1. The quality of images and text in the figures is still not optimal. For example, in figure 1, the caption at the lower part of the figure is not readable, as well as in the left part of the figure on the blue background, and etc. It is recommended to improve the resolution (pixels) of the image .

2. DiGeorge syndrome and Cornelia de Lange syndrome (CdLS) added to the article are genetic diseases, unlike the other diseases considered.  It is recommended not to add them to the article.

3. It is recommended that all abbreviations be written in full when first mentioned.

Author Response

 I am very  grateful for your time. 

This manuscript is a resubmission of an earlier submission. The following is a list of the peer review reports and author responses from that submission.